# *Ruscus aculeatus* extract promotes RNase 7 expression through ERK activation following inhibition of late-phase autophagy in primary human keratinocytes

**Shigeyuki Ono** ⓘ *, **Akiko Kawasaki, Kotaro Tamura, Yoshihiko Minegishi** ⓘ, **Takuya Mori** ⓘ, **Noriyasu Ota**

Biological Science Research, Kao Corporation, Ichikai-machi, Haga-gun, Tochigi, Japan

* ono.shigeyuki@kao.com

**Data Availability Statement:** All relevant data are within the manuscript and its Supporting Information files.

## Abstract

Antimicrobial peptides (AMPs) are crucial for protecting human skin from infection. Therefore, the expression levels of beneficial AMPs such as ribonuclease 7 (RNase 7) must be appropriately regulated in healthy human skin. However, there is limited understanding regarding the regulating AMP expression, especially when using applications directly to healthy human skin. Here, we investigated the effects of the extract of *Ruscus aculeatus* (RAE), a medicinal plant native to Mediterranean Europe and Africa that is known to have a high safety level, on AMP expression in primary human keratinocytes. Treatment with RAE induced RNase 7 expression, which was suppressed by an extracellular signal-regulated kinase (ERK) inhibitor. The autophagic flux assay and the immunofluorescence analysis of microtubule-associated protein 1 light chain 3 (LC3)-II and p62 showed that RAE inhibited late-phase autophagy. Moreover, both the inhibition of early-phase autophagy by EX-527, an inhibitor of silent information regulator of transcription 1 (SIRT1) and its enhancement by resveratrol, an activator of SIRT1 inhibited RNase 7 and ERK expression, indicating that autophagosome accumulation is necessary for RAE-induced RNase 7 expression. Additionally, spilacleoside was identified as the active component in RAE. These findings suggest that RAE promotes RNase 7 expression via ERK activation following inhibition of late-phase autophagy in primary human keratinocytes and that this mechanism is a novel method of regulation of AMP expression.

## Introduction

The human skin is constantly exposed to a variety of microorganisms. The human skin serves as a protective chemical as well as a physical barrier against infections caused by both pathogenic and opportunistic bacteria. The physical barrier is attributed to the presence of corneocytes and intercellular membrane bilayers containing ceramides. The chemical barrier is provided by the release of antimicrobial peptides (AMPs), which are produced by epidermal

**Funding:** The author(s) received no specific funding for this work.

**Competing interests:** The authors have declarted that no competing interests exist.

keratinocytes. Human β-defensin (hBD)-1, hBD-2, hBD-3, cathelicidin LL-37 (LL-37), S100 protein psoriasin (S100A7), and ribonuclease 7 (RNase 7) are well known AMPs expressed in the skin [1,2]. Among these, RNase 7 [3] and hBD-3 [4–6] are constitutively expressed in the stratum corneum, where AMPs are considered to function most effectively against infection in healthy human skin and exhibit a broad spectrum of antimicrobial activity against various gram-negative and gram-positive bacteria [3,7]. It has been reported that when traveling to tropical or subtropical areas, individuals with relatively high expression levels of RNase 7 show a higher degree of resistance to *Staphylococcus aureus* infection of the skin compared with individuals with low levels of RNase 7 expression [8]. In addition, RNase 7 inhibits the growth of dermatophytes such as *Trichophyton rubrum* and *T. mentagrophytes* in vitro [9]. In contrast, expression of hBD-3 is downregulated in the skin of patients suffering from atopic dermatitis (AD) [10]. hBD-3 increased the expression of tight junction proteins, enhanced transepithelial resistance, and reduced paracellular flux in human keratinocytes, indicating its role in reinforcing the physical barrier of skin [11]. In addition, a recent study has revealed that hBD-3 attenuates AD-like inflammation via activation of autophagy and the aryl hydrocarbon receptor signalling pathway [12]. Overall, these reports suggest that the AMPs RNase 7 and hBD-3 play important roles in cutaneous defence and maintaining constant and appropriate expression levels of these AMPs is crucial for ensuring healthy human skin.

RNase 7 expression is synergistically induced by interleukin 17A (IL-17A) and interferon-γ via signal transducer and activator of transcription 3 (STAT3) in human primary keratinocytes [13]. RNase 7 is also expressed in human primary keratinocytes by *S. aureus*, a commensal bacterium, through activation of Toll-like receptor 2, epidermal growth factor receptor, and nuclear factor κB [14], and by *T. rubrum*, a dermatophyte, in an epidermal growth factor-dependent manner [15]. In contrast, hBD-3 expression in human primary keratinocytes is known to be induced by proinflammatory cytokines such as tumour necrosis factor-α (TNF-α) [16], infections caused by bacteria such as *Pseudomonas aeruginosa* and *S. aureus* [16], and ceramide-1-phosphate produced following endoplasmic reticulum stress [17]. These studies imply that the expression levels of RNase 7 and hBD-3 are upregulated under inflammatory conditions or by inflammatory factors such as cytokines. However, it would be difficult to directly employ bacteria-evoked inflammatory conditions or inflammatory factors such as cytokines as a means which modulate the expression levels of beneficial AMPs such as RNase 7 and hBD-3 to healthy human skin. Therefore, in order to constantly regulate and maintain the expression levels of these AMPs at appropriate levels in healthy human skin, it is crucial to develop means that can be readily used for healthy human skin: means that does not utilize inflammatory conditions or inflammatory factors.

The main aim of our study is to explore a means of regulating the expression levels of beneficial AMPs that does not directly apply inflammatory conditions or inflammatory factors to healthy human skin. In the course of our studies on bioactive plant extracts, we found that the extract from *Ruscus aculeatus* (RA) promotes RNase 7 expression in primary human keratinocytes. RA, also known as butcher's broom, is a medicinal plant native to Mediterranean Europe and Africa. RA extract (RAE) is mainly prepared from the roots of RA and shows medicinal effects against venous insufficiency, oedema, premenstrual syndrome, and haemorrhoids [18]. RAE is generally considered to be a safe medicine which is also used as a cosmetic ingredient [19,20]. In the present study, we investigated the effect of RAE on AMP expression and the underlying molecular mechanisms, using primary human keratinocytes. We found that RAE promotes the expression of AMPs, in particular RNase 7, through ERK activation following the inhibition of late-phase autophagy in primary human keratinocytes, indicating that autophagosome accumulation is important for RAE-induced RNase 7 expression. This study also proposes a novel role for autophagy in human skin.

## Materials and methods

### Reagents

Normal human epidermal keratinocytes (NHEKs) were obtained from Thermo Fisher Scientific (Waltham, MA, USA). Hydroxychloroquine sulfate (HCQ), bafilomycin A1 (BA1), PD98059, wortmannin, EX-527, and resveratrol were purchased from Fujifilm Wako Pure Chemical Corporation (Osaka, Japan). The following primary antibodies and dilution concentrations were used for western blotting in this study: anti-ribonuclease 7 (mouse, 1/1,000; Abcam, ab154143), anti-phospho p44/42 MAPK (Erk1/2; Thr202/Tyr204; rabbit, 1/2,000; Cell Signaling Technology, #4370), anti-p44/42 MAPK (Erk1/2; rabbit, 1/2,000; Cell Signaling Technology, #4695), anti-phospho SAPK/JNK (Thr183/Tyr185; rabbit, 1/2,000; Cell Signaling Technology, #4668), anti-SAPK/JNK (rabbit, 1/2,000; Cell Signaling Technology, #9252), anti-LC3B (rabbit, 1/2,000; Cell Signaling Technology, #2775), anti-SQSTM1/p62 (rabbit, 1/2,000; Cell Signaling Technology, #5114), anti-phospho S6 ribosomal protein (Ser235/236; rabbit, 1/2,000; Cell Signaling Technology, #4858), anti-S6 ribosomal protein (rabbit, 1/2,000; Cell Signaling Technology, #2217), anti-β-actin (rabbit, 1/5,000; Cell Signaling Technology, #4970), and anti-β-actin (mouse, 1/5,000; Sigma-Aldrich, St. Louis, MO, USA; A5441). The secondary antibodies used for western blotting were horseradish peroxidase (HRP)-conjugated rabbit IgG (1/2,000 or 1/5,000; Cell Signaling Technology, #7074) and HRP-conjugated mouse IgG (1/2,000 or 1/5,000; Cell Signaling Technology, #7076). The primary antibodies used for immunofluorescence staining were anti-LC3 (rabbit, 1/1,000; MBL, PM036) and anti-SQSTM1/p62 (mouse, 1/1,000; Cell Signaling Technology, #88588). The secondary antibodies used for immunofluorescence staining were as follows: donkey anti-rabbit IgG (H + L) highly cross-adsorbed secondary antibodies Alexa Fluor 488 (1/1,000; Thermo Fisher Scientific, A-21206) and goat anti-mouse IgG (H + L) highly cross-adsorbed secondary antibodies Alexa Fluor 647 (1/1,000; Thermo Fisher Scientific, A-21236).

### Preparation of *R. aculeatus* extract

Dried rhizomes from Republic of Albania were obtained from Ichimaru Pharcos Company Limited (Gifu, Japan). The extracts were prepared by submerging dried rhizomes of *R. aculeatus* (cut into small pieces; total weight: 15 kg) in ethanol (30%; 150 L) for a week at room temperature. Subsequently, the extract was filtered, and the filtrate was adsorbed on DIAION HP20 (20 L; Mitsubishi Chemical Corporation, Tokyo, Japan) and washed with ethanol (50%; 60 L). The adsorbates were eluted with ethanol (99.5%; 60 L). The eluent was stirred with an appropriate amount of activated carbon for 3 h at room temperature. After filtration, the eluent was concentrated using a rotary evaporator (Tokyo Rikakikai Co., Ltd., Tokyo, Japan) and lyophilised with a freeze-dryer (Tokyo Rikakikai Co., Ltd.). As a result, a solid substance (250 g; concentrate of *R. aculeatus*) was obtained.

The *R. aculeatus* extract (RAE) was prepared by dissolving the solid substance in ethanol (30%; solid substance dissolved at a concentration of 1% [w/v]).

### Isolation of active compound from *R. aculeatus* extract

The obtained solid substance (2 g) was subjected to column chromatography (Universal Columns Premium, Yamazen Corporation, Osaka, Japan) using chloroform to methanol as the gradient mobile phase. This process was repeated three times, and a fraction (54.2 mg) containing the active compound was obtained. Furthermore, 5.2 mg of the above fraction was subjected to preparative HPLC (L-column2 ODS; 10 × 250 mm; Chemical Evaluation and Research Institute, Tokyo, Japan) using formic acid (0.1%) to acetonitrile as the gradient

mobile phase. Finally, spilacleoside (1.9 mg), the active compound present in RAE, was isolated (yield: 0.99% from the solid substance obtained from *R. aculeatus*).

## Cell culture and treatment

NHEKs were cultured at 37°C under $CO_2$ (5%) in Epilife medium (Thermo Fisher Scientific) containing insulin (10 μg/mL), human recombinant epidermal growth factor (0.1 μg/mL), hydrocortisone (0.5 μg/mL), gentamycin (50 μg/mL), amphotericin B (50 ng/mL), and bovine pituitary extract (0.4% [v/v]) (KK-6150, Kurabo Industries, Osaka, Japan). NHEKs were seeded in 6-well plates (density: $1.0 \times 10^5$ cells per well) and cultured until approximately 80% confluency. The medium was subsequently replaced with a medium that did not contain human recombinant epidermal growth factor and bovine pituitary extract and cultured for an additional 24 h. The cells were then incubated with RAE and/or the indicated reagents for the indicated times. RAE diluted with ethanol (30%), HCQ dissolved in water, and BA1, PD98059, wortmannin, EX-527, and resveratrol dissolved in dimethyl sulfoxide (DMSO; Fujifilm Wako Pure Chemical Corporation) were used at the indicated concentrations. The maximum concentration of each solvent in each medium was kept below 0.1% (v/v).

## MTT assay

NHEKs were seeded in 12-well plates (density: $1.0 \times 10^5$ cells per well). Cells were incubated with or without RAE (0.1% [v/v]) for 72 h. Then, the medium was removed, and the cells were incubated with MTT solution (final concentration: 0.5 mg/mL) for 3 h at 37°C under $CO_2$ (5%). Next, DMSO (1 mL) was added to extract the formazan crystals, and 100 μL of the solution was transferred to a 96-well plate. The absorbance of each well was measured at 570 nm using an automated microplate reader (PowerWave XS, BioTek Instruments, Winooski, VT, USA).

## Autophagic flux assay

NHEKs were seeded in 6-well plates (density: $1.0 \times 10^5$ cells per well). The cells were treated with RAE (0.1% [v/v]) for 72 h. Next, the cells were treated with HCQ (10 μM) and BA1 (50 nM) 24 h before harvesting and were subjected to western blotting. The protein expression levels of LC3-II induced by RAE in the presence of HCQ were compared with those induced by RAE alone.

## Real-time quantitative PCR (RT-qPCR)

Total RNA was extracted from keratinocytes using the RNeasy Plus Mini Kit (Qiagen, Tokyo, Japan) or Maxwell 16 LEV simplyRNA Tissue Kit (Promega, Tokyo, Japan).

Complementary DNA was synthesised from total RNA (1 μg) using a high-capacity RNA-to-cDNA kit (Thermo Fisher Scientific). RT-qPCR was performed using the TaqMan Universal PCR Master Mix (Thermo Fisher Scientific). Amplification and detection of mRNAs of all AMPs were performed using a real-time PCR system (model number 7500, Thermo Fisher Scientific). The assay IDs of the primer/probe sets (Thermo Fisher Scientific) used were as follows: Hs00922963_s1 for RNase 7, Hs00189038_m1 for LL-37, Hs00608345_m1 for hBD-1, Hs00175474_m1 for hBD-2, Hs00218678_m1 for hBD-3, and Hs00161488-m1 for S100A7. To normalise mRNA concentrations, transcript levels of the housekeeping gene ribosomal protein lateral stalk subunit P0 (RPLP0) were determined for each sample (assay ID Hs99999902_m1), and relative transcript levels of each sample were corrected based on the RPLP0 transcript levels. Changes in gene expression are indicated as fold-increases relative to untreated controls.

## Western blotting

After stimulation, the supernatant was removed. The cells were washed twice with PBS and lysed in RIPA buffer (220 μL; Sigma-Aldrich) supplemented with a complete protease inhibitor tablet (1 tablet per 10 mL; Roche, Mannheim, Germany). The lysates were centrifuged at 15,000 rpm for 10 min at 4˚C, and the supernatants were collected. The BCA protein assay (Pierce Biotechnology, Rockford, IL, USA) was used for protein quantification. The samples were mixed with 4× Laemmli Sample Buffer (Bio-Rad Laboratories, Hercules, CA, USA), RIPA buffer (Sigma-Aldrich), and DL-dithiothreitol solution (10% v/v; Sigma-Aldrich) and then boiled for 5 min. Equal amounts of each sample (5 or 10 μg) were loaded onto Mini-PRO-TEAN TGX Gels (4–20%; Bio-Rad Laboratories), and SDS-PAGE was performed at 50–80 V. The proteins were transferred onto activated PVDF membrane (Bio-Rad Laboratories) at 100 V for 60 min or using Trans-Blot Turbo Transfer System (Bio-Rad Laboratories). The membrane was immersed in PVDF Blocking Reagent (TOYOBO, Osaka, Japan) for 1 h at room temperature to block nonspecific binding sites, followed by overnight incubation at 4˚C in Can Get Signal Solution 1 (TOYOBO) containing specific primary antibody at an appropriate dilution. After washing three times with TBS-T, the membrane was incubated at room temperature for 1 h in Can Get Signal Solution 2 (TOYOBO) containing secondary antibody conjugated to HRP at an appropriate dilution. The membrane was then washed with TBS-T three times and developed with enhanced chemiluminescence (ECL) detection reagent (Cytiva, Tokyo, Japan). Visualisation and density measurements of the blots were performed using a ChemiDoc MP Imaging System (Bio-Rad Laboratories). Blot densities of phosphorylated ERK, JNK, and S6 were normalised according to the blot densities of total ERK, JNK, and S6, respectively. In addition, the blot densities of the other proteins were normalised against the blot density of β-actin. Changes in protein expression levels are indicated as fold-increases relative to untreated controls.

## Immunofluorescence staining

Sterile coverslips were coated with type I collagen (14 mm diameter, AGC Techno Glass Co., Ltd., Shizuoka, Japan) and placed in 12-well plates. On these coverslips, NHEKs were seeded (density: $1.0 \times 10^5$ cells per well). The cells were incubated with or without RAE (0.1% [v/v]) for 72 h. Then, the medium was removed, and the cells were fixed with 4% paraformaldehyde (Fujifilm Wako Pure Chemical Corporation) for 10 min, washed with PBS three times, permeabilised with 50 μg/mL digitonin (Fujifilm Wako Pure Chemical Corporation) in PBS, blocked with 0.1% gelatin in PBS, and incubated with specific primary antibody at an appropriate dilution for 60 min at room temperature. After washing three times with PBS, the cells were incubated with secondary antibody at an appropriate dilution for 40 min at room temperature. The cells were then washed with PBS three times and mounted using Vectashield Mounting Medium with DAPI (H-1200, Vector Laboratories, Newark, CA, USA).

Images of the stained cells were acquired using an all-in-one fluorescence microscope (BZ-X700, Keyence, Osaka, Japan) with Z-stacks at 10–15 μm depth from the surface and 1 μm intervals at a magnification of 100×. Quantification of dots derived from LC3, p62, and colocalisation was performed using a hybrid cell counting system (BZ Analyser; Keyence). Three field images per coverslip were randomly acquired (approximately 80–90 cells), and five experiments were analysed for each condition. The dots were automatically quantified under similar conditions using the macro cell count mode. The total number of cells analysed per condition was approximately 400–450.

### Statistical analysis

Data are presented as means ± standard deviation (SD). Statistical analysis was performed using KyPlot 5.0 (Kyenslab Inc., Tokyo, Japan). Differences between two groups were compared using Student's $t$-test. For groups more than two, differences were compared by one-way analysis of variance (ANOVA) followed by post-hoc Tukey's multiple comparison test. Statistical significance was set at $P < 0.05$.

For the analysis of differences between control and RAE, control and EX (EX-527), and control and RAE + EX (Fig 4G), Bonferroni correction was performed. Thus, a P-value $< 0.0166$ (0.05/3) was considered statistically significant.

## Results

### *R. aculeatus* extract promotes expression of ribonuclease 7

First, we investigated the effect of RAE on mRNA expression levels of RNase 7 in primary human keratinocytes. RNase 7 is considered to be an important AMP expressed and functional in the stratum corneum of human skin. RAE significantly increased the mRNA expression levels of RNase 7 in a concentration- and time-dependent manner compared with control (Fig 1A and 1B). Moreover, 3-[4,5-dimethylthiazol-2-yl]-2,5-diphenyltetrazolium bromide (MTT)

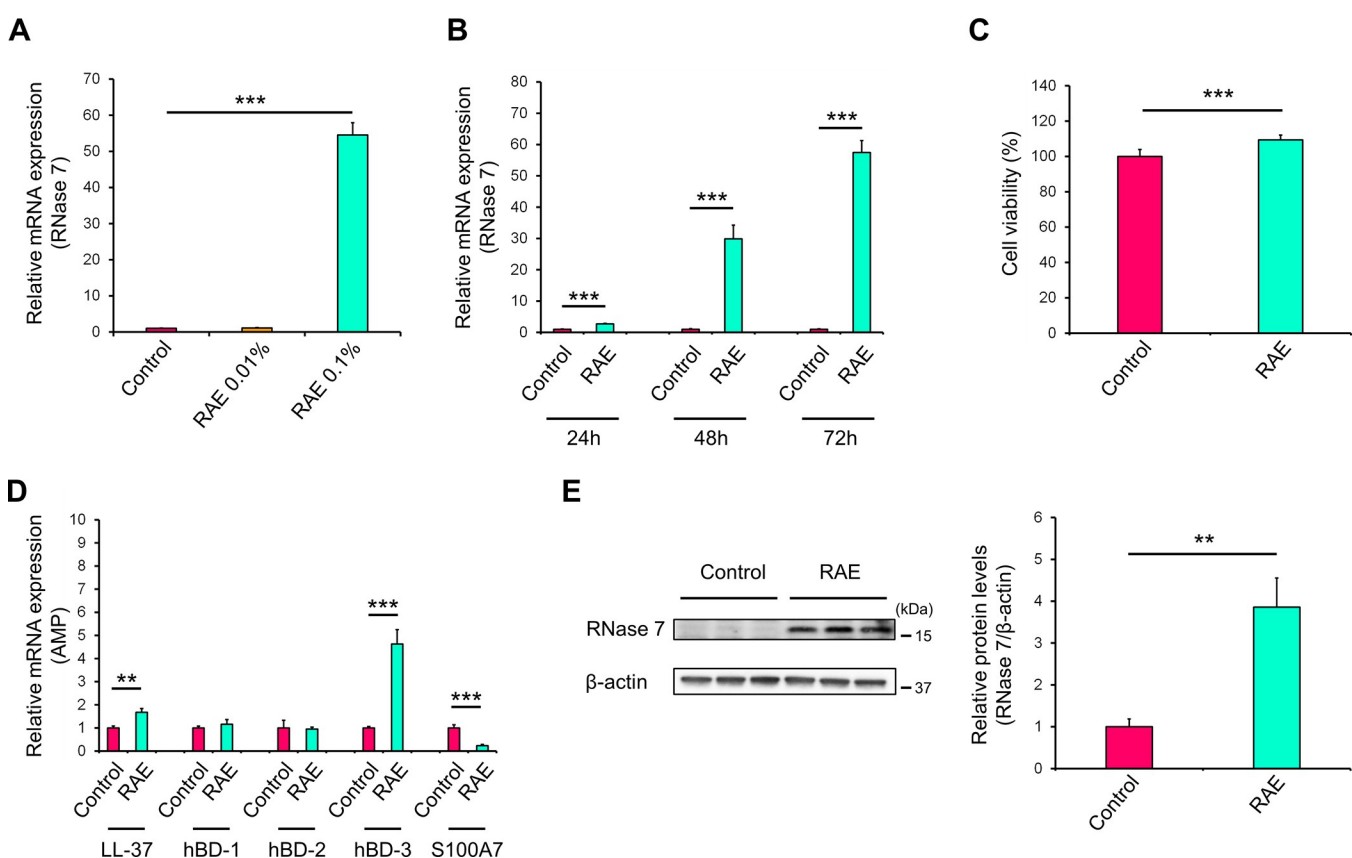

**Fig 1. Effect of Ruscus aculeatus extract (RAE) on antimicrobial peptide (AMP) expression in primary human keratinocytes.** (**A–E**) Normal human epidermal keratinocytes (NHEKs) were treated with RAE (0.1% [v/v]) for 72 h. (**A**) Effect of RAE on mRNA expression levels of ribonuclease 7 (RNase 7) was determined by RT-qPCR. (**B**) Time-dependency of RAE-induced RNase 7 expression was analysed by examining mRNA expression levels of RNase 7 via RT-qPCR. (**C**) Effect of RAE on cell viability was measured using MTT assay. (**D**) Influence of RAE on mRNA expression levels of various AMPs was determined by RT-qPCR. (**E**) Protein expression levels of RNase 7 were analysed by western blotting. (**A–E**) Data represent the mean ± SD of three independent experiments. (**A**) One -way ANOVA with Tukey's test. ***P < 0.001. (**B–E**) Student's t-test. ** P < 0.01, ***P < 0.001.

assay of primary human keratinocytes treated with RAE (0.1% [v/v]) for 72 h showed that the viability of RAE-treated cells was significantly higher than control (Fig 1C). This indicated that treatment with RAE did not injure the cells. Thus, subsequent experiments were performed using cells treated with 0.1% [v/v] RAE for 72 h.

Next, we assessed whether RAE treatment influenced the mRNA expression levels of AMPs other than RNase 7. RAE treatment significantly increased hBD-3 and LL-37 expression levels compared with control (Fig 1D). However, the mRNA expression levels of hBD-3, especially LL-37, were not high as those of RNase 7. The effect of RAE on the protein expression levels of RNase 7 was investigated using western blot analysis. RNase 7 protein expression was significantly increased following RAE treatment (Fig 1E). In summary, our findings demonstrated that RAE promoted RNase 7 expression. Therefore, further experiments were performed to elucidate the mechanisms of induction of AMP expression, especially RNase 7 expression, by RAE.

## Extracellular signal-regulated kinase activation is involved in *R. aculeatus* extract-induced ribonuclease 7 expression

Mitogen-activated protein kinases (MAPKs) are involved in AMP expression in primary human keratinocytes [21]. Therefore, the activation levels of ERK and c-Jun N-terminal kinase (JNK) were evaluated in this study. RAE significantly increased the phosphorylation levels of ERK and JNK protein compared with control (Fig 2A and 2B). However, the activation of JNK was considered negligible, as it was weaker than that of ERK, indicating that RAE primarily activates ERK. To investigate the involvement of ERK signalling in RAE-induced RNase 7 expression, the cells were treated with RAE in the presence of PD98059, a mitogen-activated protein kinase kinase (MEK)/ERK inhibitor. The MEK/ERK inhibitor significantly diminished the REA-induced increase in the mRNA expression levels of RNase 7 (Fig 2C; RAE vs. RAE+PD98059). Moreover, the increase in the protein expression levels of phosphorylated ERK and RNase 7 caused by RAE was significantly suppressed in the presence of PD98059 (Fig 2D; RAE vs. RAE+PD and 2E; RAE vs. RAE+PD). These findings suggest that ERK activation is involved in RAE-induced RNase 7 expression.

## *R. aculeatus* extract inhibits late-phase autophagy

1,25-dihydroxyvitamin $D_3$ is well known to induce LL-37 expression in human skin [22]. In contrast, it has been reported that calcipotriol, a vitamin $D_3$ analogue, induces autophagy in HeLa cells and human keratinocytes [23]. These reports suggest that AMP expression may influence autophagy. Therefore, we examined the association between RAE-induced RNase 7 expression and autophagy in primary human keratinocytes.

Microtubule-associated protein 1 light chain 3 (LC3) is a typical autophagy marker [24]. RAE treatment significantly increased the protein levels of LC3-II compared with control (Fig 3A), suggesting that RAE affects autophagy. However, this result was insufficient to determine whether autophagy was induced or inhibited by RAE [25]. To address this issue, an autophagic flux assay [25] was performed using hydroxychloroquine (HCQ) and bafilomycin A1 (BA1), potent autophagy inhibitors. HCQ and BA1 inhibit the fusion of autophagosome with lysosome; in other words, they inhibit late-phase autophagy (the process where autophagosome fuses with lysosome, autolysosome is formed, and cargos are degraded to their components, such as amino acids). In the presence of HCQ (Fig 3B; RAE vs. RAE+HCQ) or BA1 (Fig 3C; RAE vs. RAE+BA1), the increase in LC3-II protein expression levels induced by RAE did not exceed the increase observed with RAE treatment alone, indicating that RAE treatment inhibits late-phase autophagy in primary human keratinocytes.

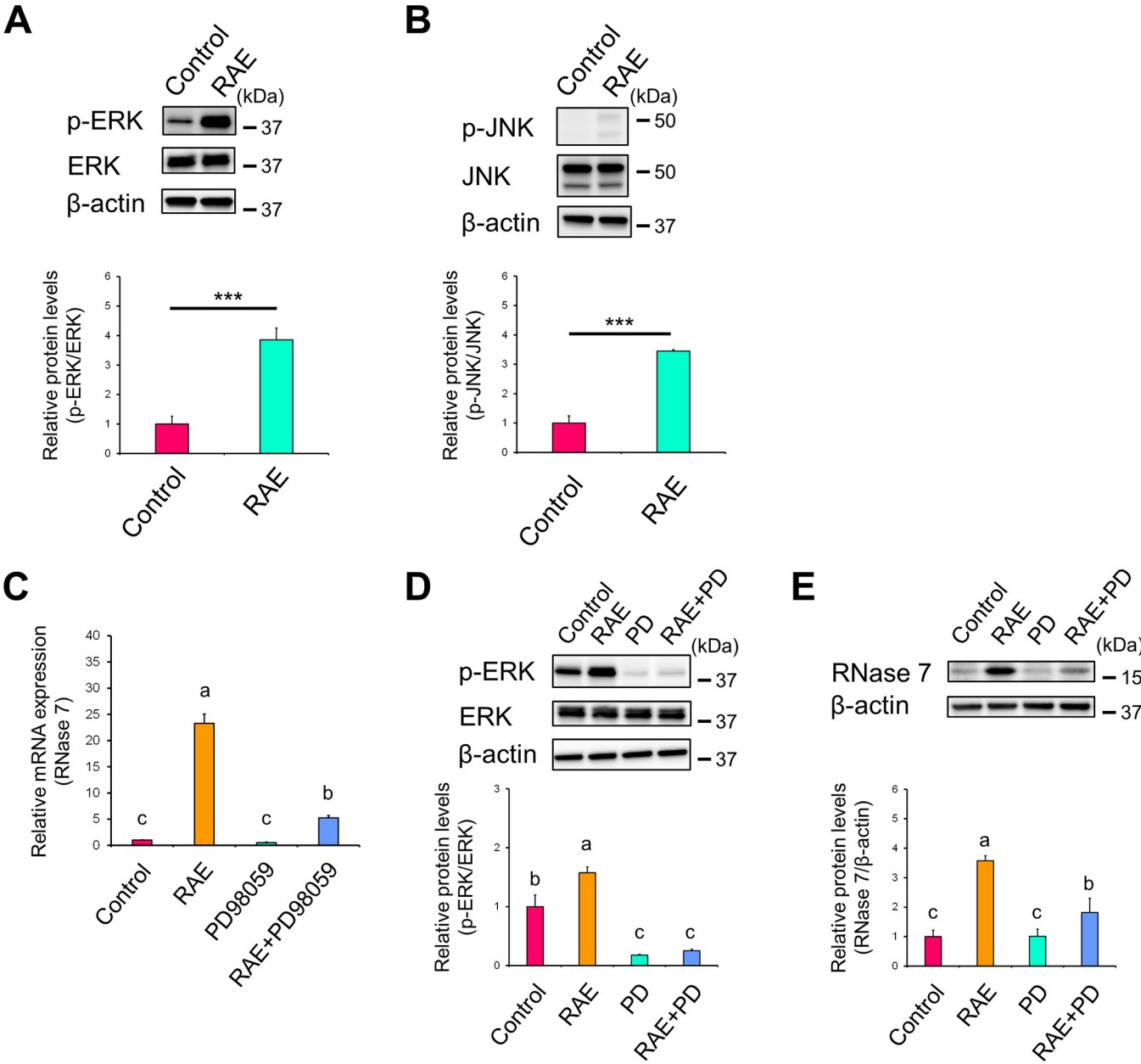

**Fig 2. Relationship between RAE-induced RNase 7 expression and MAPKs.** (**A, B**) NHEKs were treated with RAE (0.1% [v/v]) for 72 h. (**A, B**) Phosphorylation levels of ERK (A) and JNK (B) proteins were determined by western blotting. (**C–E**) NHEKs were treated with RAE (0.1% [v/v]) for 72 h in the presence of PD98059, a MEK/ERK inhibitor (30 μM). (**C**) mRNA expression levels of RNase 7 were analysed by RT-qPCR. (**D, E**) Protein expression levels of phosphorylated ERK (D), and RNase 7 (E) were determined by western blotting. (**A–E**) Data represent the mean ± SD of three independent experiments. (**A, B**) Student's *t*-test. ** $P < 0.01$, ***$P < 0.001$. (**C–D**) Different letters indicate significant differences based on post-hoc Tukey's test results. $P < 0.05$. PD: PD98059.

The protein p62, also known as sequestosome (SQSTM), is another common marker of autophagy, as its levels increases when late-phase autophagy is inhibited [26]. Ubiquitinated cargos (misfolded proteins, protein aggregates, damaged mitochondria, bacteria, and viruses) interact with p62 and are subsequently sequestered in autophagosomes via interaction with LC3-II on the inner membrane [27,28]. Since impaired autophagy causes the accumulation of autophagosomes, the number of colocalised dots of LC3 and p62 increased in

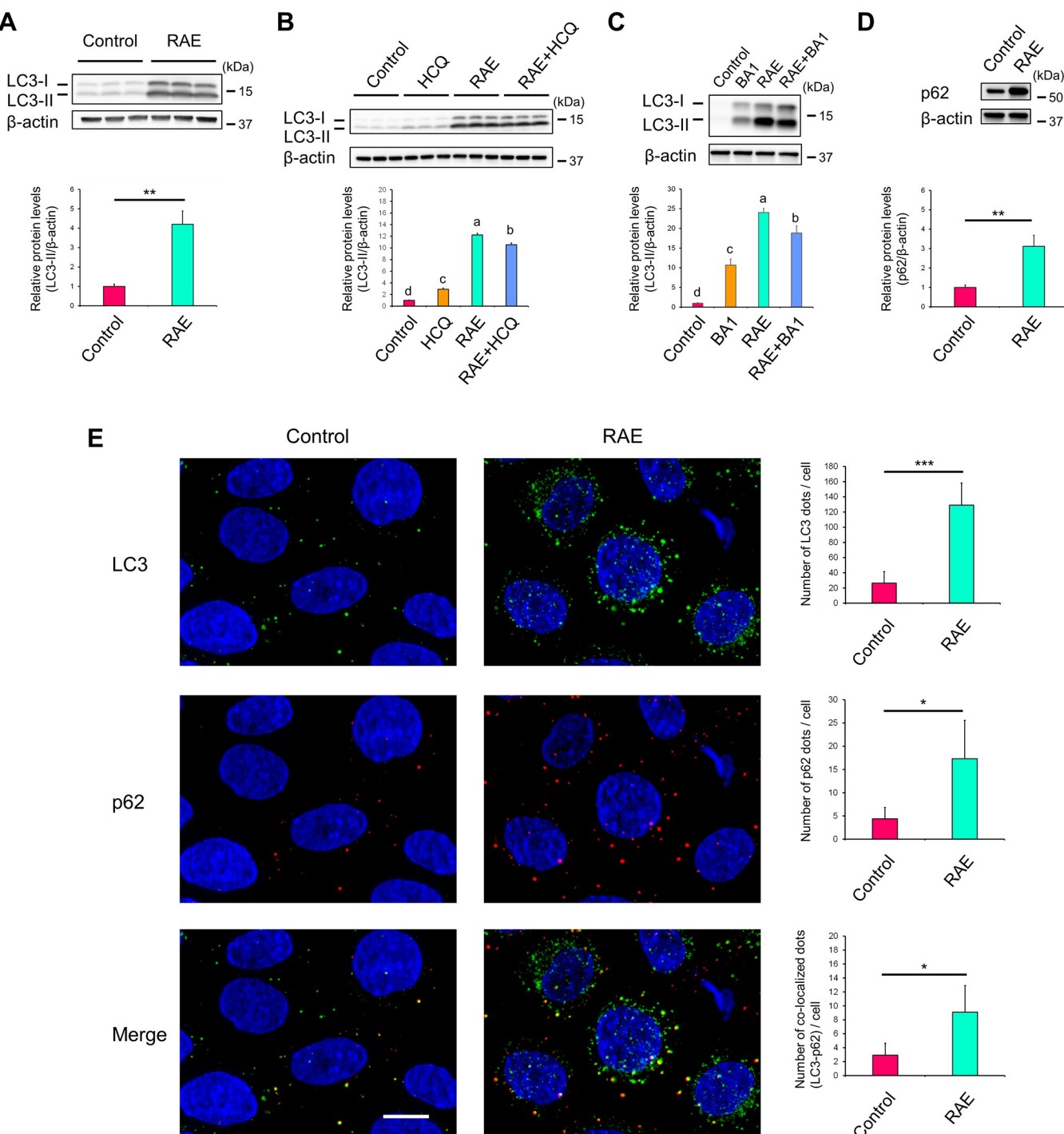

**Fig 3. Late-phase autophagy is inhibited by RAE.** (**A–D**) NHEKs were treated with RAE (0.1% [v/v]) for 72 h. (**A**) Protein expression levels of LC3-II, an autophagy marker, were determined by western blotting. (**B, C**) Autophagic flux assay was performed using HCQ (B) or BA1 (C). HCQ (10 μM) or BA1 (50 nM) was added into the medium 24 h before harvest. (**D**) Protein expression levels of p62, an autophagy marker, were determined by western blotting. (**E**) Colocalisation of LC3 and p62 was analysed by immunofluorescence staining. (**A–E**) Data represent the mean ± SD of three independent experiments. (**A, D, E**) Student's *t*-test. * P < 0.05, ** P < 0.01, ***P < 0.001. (**B, C**) Different letters indicate significant differences based on post-hoc Tukey's test results. P < 0.05. (**E**) Scale bar, 10 μm. HCQ: Hydroxychloroquine sulfate, BA1: Bafilomycin A1.

immunofluorescence analysis [29]. Consequently, to further confirm the inhibition of late-phase autophagy by RAE treatment, we examined the protein expression levels of p62 and colocalisation of LC3 and p62 dots. RAE treatment significantly increased the protein expression levels of p62 (Fig 3D) and the number of colocalised dots of LC3 and p62 (Fig 3E) compared with control. These results also support the notion that RAE treatment inhibits late-phase autophagy and induces the accumulation of autophagosomes in primary human keratinocytes.

## Early-phase autophagy is required for *R. aculeatus* extract-induced ribonuclease 7 expression

As shown above, RAE inhibited late-phase autophagy. However, it is unclear whether early-phase autophagy (the process where the isolation membrane occurs, unwanted materials such as misfolded proteins are incorporated into it, the opening of the membrane closes, and finally autophagosome is formed), i.e., basal autophagic activity, is essential for RAE-induced RNase 7 expression. Therefore, the association of early-phase autophagy with RAE-induced RNase 7 expression was examined using wortmannin, a typical basal autophagic inhibitor that inhibits phosphoinositide 3-kinases (PI3Ks), which are involved in the mammalian target of rapamycin (mTOR)-dependent autophagic pathway [30]. In the presence of wortmannin, the RAE-induced increase in mRNA expression levels of RNase 7 was significantly inhibited (Fig 4A; RAE vs. RAE+Wortmannin), indicating that early-phase autophagy based on mTOR-dependent autophagy pathway participates in RAE-induced RNase 7 expression.

SIRT1 is an important regulator of autophagy, including basal levels of autophagy [31]. Early-phase autophagy can be suppressed by mTOR activation [32]. Moreover, SIRT1 was demonstrated to negatively regulate mTOR signalling pathway, potentially through tuberous sclerosis complex 1/2 (TSC1/2) complex [33]. These studies suggest that SIRT1 inhibitors, which activate mTOR, can suppress early-phase autophagy. Indeed, the SIRT1 inhibitor nicotinamide increased the phosphorylation levels of ribosomal protein S6 (S6), a downstream target of mTOR [33]. In addition, numerous reports have shown that EX-527 inhibits autophagy via inhibition of the SIRT1 receptor [34–36]. Therefore, using EX-527, a SIRT1 inhibitor, we more directly examined the participation of early-phase autophagy in RAE-induced RNase 7 expression. EX-527 effectively inhibited the RAE-induced increase in mRNA expression levels of RNase 7 (Fig 4B; RAE vs. RAE+EX-527). This result also suggests that early-phase autophagy based on mTOR-dependent autophagy pathway is involved in RAE-induced RNase 7 expression.

Since a SIRT1 inhibitor is considered appropriate to study the participation of early-phase autophagy in RAE-induced RNase 7 expression, we next investigated the influence of EX-527 on protein expression levels of RNase 7, phosphorylated ERK, LC3, and p62. In the presence of EX-527, the RAE-induced increase in protein expression levels of RNase 7 (Fig 4C; RAE vs. RAE+EX) and phosphorylated ERK (Fig 4D; RAE vs. RAE+EX) was significantly diminished compared with RAE alone. These findings also support the notion that early-phase autophagy participates in RAE-induced RNase 7 expression. However, the protein expression levels of LC3-II (Fig 4E; RAE vs. RAE+EX) and p62 (Fig 4F; RAE vs. RAE+EX) did not change in the presence of EX-527. In general, when early-phase autophagy is suppressed, it is expected that the protein expression levels of LC3-II and p62 were altered (e.g., reduced for LC3-II, increased for p62). Thus, these results contradict the expected results associated with the suppression of early-phase autophagy.

The SIRT1 inhibitor, which activates mTOR and suppresses early-phase autophagy, has been reported to activate S6, a downstream target of mTOR [33]. Namely, when early-phase

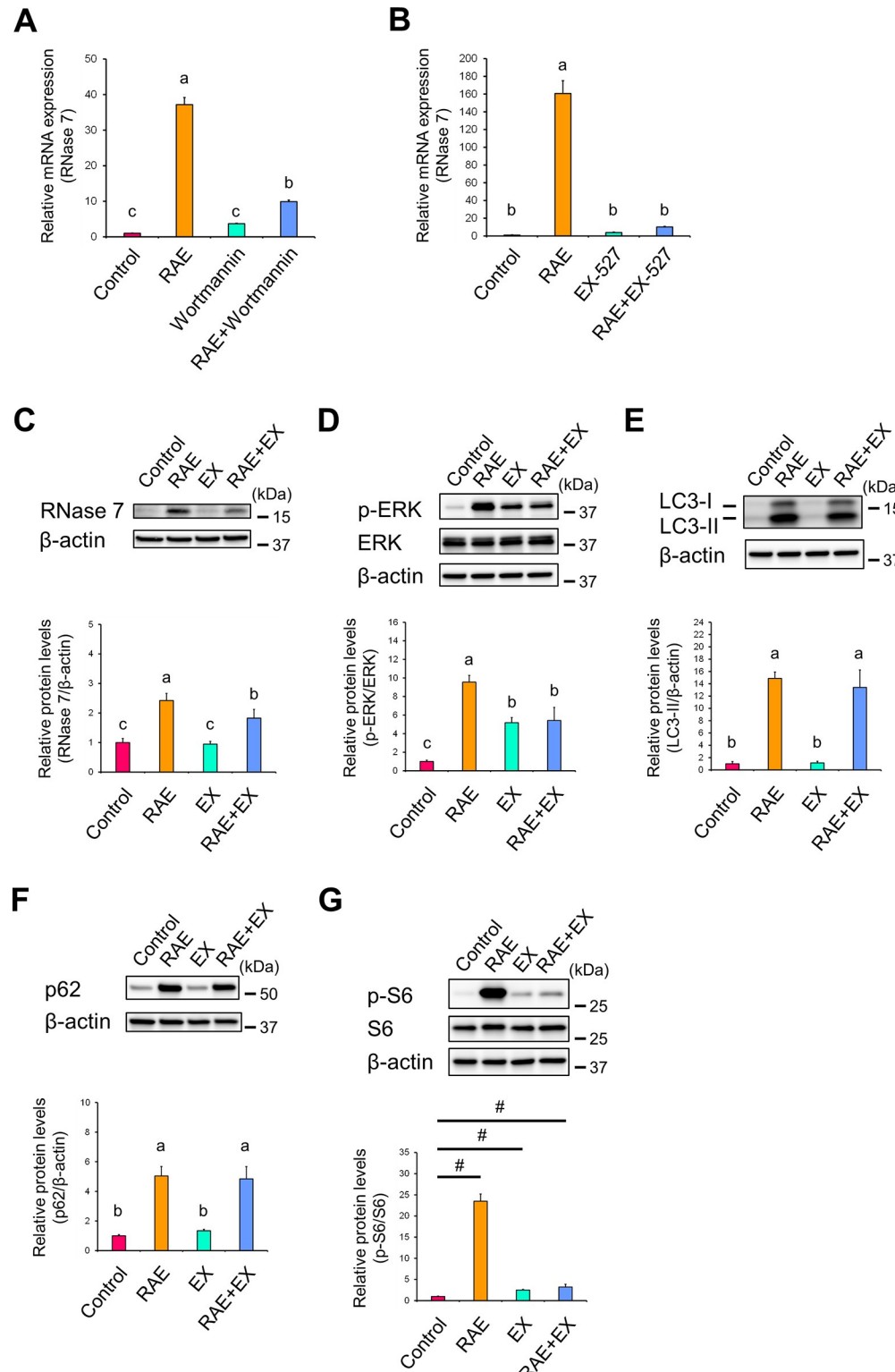

**Fig 4. RAE-induced RNase 7 expression requires basal autophagic activity.** (**A–G**) NHEKs were treated with RAE (0.1% [v/v]) for 72 h in the presence of wortmannin (10 μM), a PI3K inhibitor (A) or EX-527 (30 μM), a SIRT1 inhibitor (B–G). (**A, B**) mRNA expression levels of RNase 7 were determined by RT-qPCR. (**C–G**) Protein expression levels of RNase 7 (C), phosphorylated ERK (D), LC3-II (E), p62 (F), and phosphorylated S6 (G) were analysed by western blotting. (**A–G**) Data represent the mean ± SD of three independent experiments. (**A–F**) Different letters indicate significant differences based on post-hoc Tukey's test results. P < 0.05. (**G**) Differences in results of control versus RAE, control versus EX, and control versus RAE + EX were analysed by Bonferroni correction. #P < 0.0166 (0.05/3). EX: EX-527.

autophagy is suppressed by EX-527, this report indicates that the protein expression levels of phosphorylated S6 in the presence of EX-527 increase compared with those in the absence of EX-527 (control). Therefore, to confirm that early-phase autophagy was suppressed by EX-527, the protein expression levels of phosphorylated S6 were examined under conditions of RAE-induced RNase 7 expression. The protein expression levels of phosphorylated S6 following treatment with EX-527 alone (Fig 4G; Control vs. EX) and the combination of RAE and EX-527 (Fig 4G; Control vs. RAE+EX) were significantly increased compared with control. Notably, treatment with RAE alone caused an unexpected and prominent increase in the protein expression levels of phosphorylated S6 compared with the other treatments (Fig 4G; RAE vs. Control, EX, or RAE+EX), which resulted in errors in the statistical analysis. Therefore, we used the Bonferroni correction for this analysis. These results implied that EX-527 suppressed early-phase autophagy under our experimental conditions.

Taken together, we concluded that early-phase autophagy (basal autophagic activity), which participates in the formation of autophagosomes, is required for RNase 7 expression that is induced following RAE treatment.

## Inhibition of late-phase autophagy is necessary for *R. aculeatus* extract-induced ribonuclease 7 expression

We observed that RAE inhibited late-phase autophagy and autophagosomes were accumulated by RAE treatment. However, it remains unclear whether the inhibition of late-phase autophagy by RAE (i.e., accumulation of autophagosomes) participated in RNase 7 expression. If it does, even in the presence of RAE, an inhibitor of late-phase autophagy, if early-phase autophagy is promoted beyond its original activity by an autophagy inducer, early-phase autophagy will exceed the inhibitory effect of RAE on late-phase autophagy. This implies that autophagy goes from an unprogressive state to a progressive state. Therefore, since autophagosomes are degraded and do not accumulate, RAE-induced RNase 7 expression in the presence of autophagy inducers is expected to be smaller than that of RAE alone. We demonstrated that early-phase autophagy, regulated by SIRT1, is involved in RAE-induced RNase 7 expression, and SIRT1 activators such as resveratrol have been reported to induce autophagy [37]. Therefore, using resveratrol as an autophagy inducer, we investigated whether inhibition of late-phase autophagy is required for RNase 7 expression following RAE treatment. Resveratrol significantly reduced the RAE-induced increase in mRNA expression levels of RNase 7 (Fig 5A; RAE vs. RAE+Resveratrol), suggesting that inhibition of late-phase autophagy is required for RAE-induced RNase 7 expression.

Similar to the experiment with EX-527, we examined the effect of resveratrol on the protein expression levels of RNase 7, phosphorylated ERK, LC3, and p62. Resveratrol treatment significantly diminished the protein expression levels of RNase 7 (Fig 5B; RAE vs. RAE+Res) and phosphorylated ERK (Fig 5C; RAE vs. RAE+Res) compared with RAE alone, suggesting that inhibition of late-phase autophagy by RAE is required for RAE-induced RNase 7 expression. However, as similar to EX-527, LC3-II (Fig 5D; RAE vs. RAE+Res) and p62 protein levels (Fig 5E; RAE vs. RAE+Res) were not influenced by the presence of resveratrol, which was not consistent with the results that were expected when early-phase autophagy was accelerated.

In contrast to treatment with EX-527, resveratrol treatment, which induces autophagy (accelerates early-phase autophagy), inactivates S6, a downstream target of mTOR [33]. Namely, when early-phase autophagy is accelerated by resveratrol, the protein expression levels of phosphorylated S6 in the presence of resveratrol decrease compared with those in the absence of resveratrol (control). To confirm that under our experimental conditions, early-phase autophagy was accelerated by resveratrol, the protein expression levels of

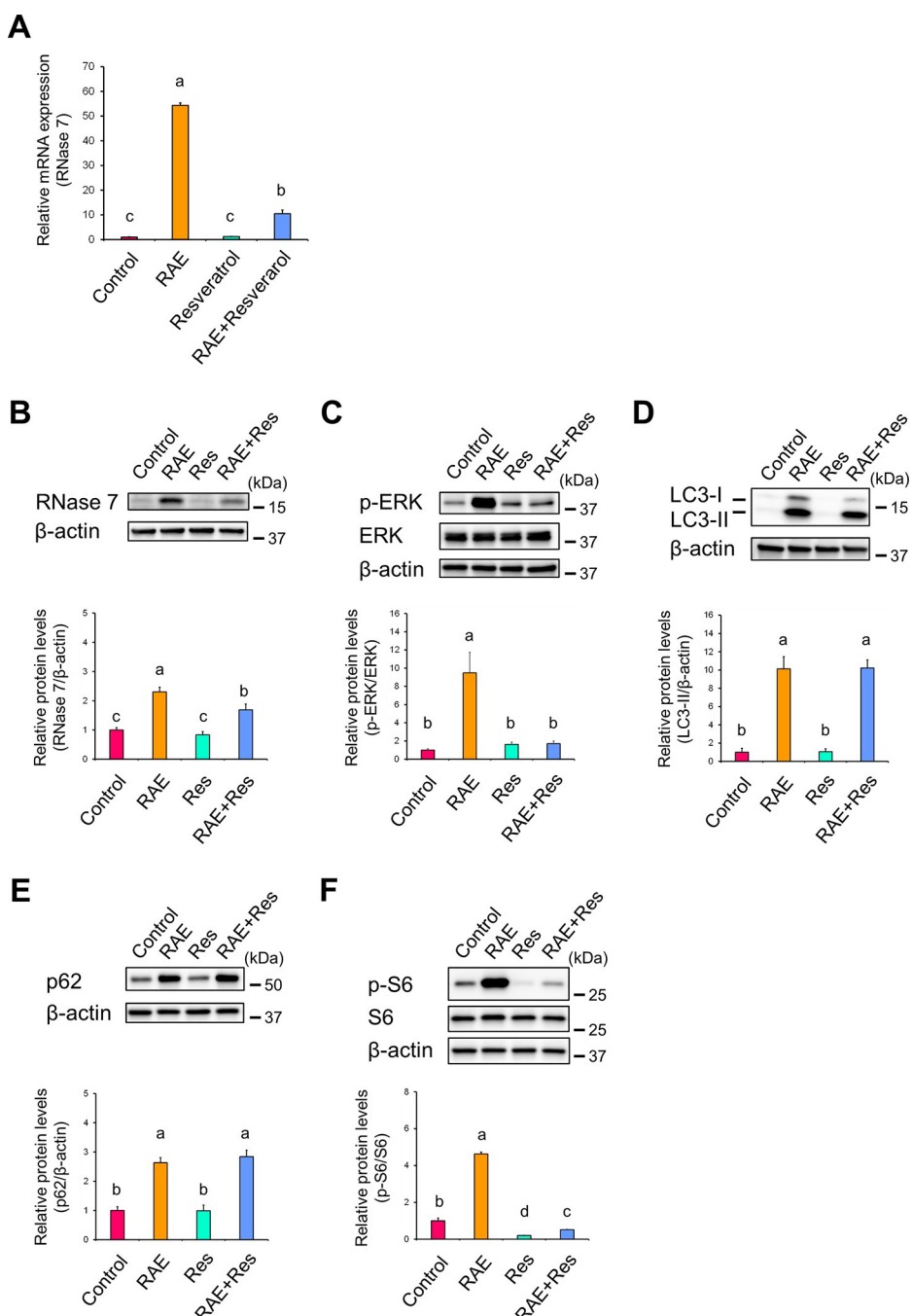

**Fig 5. RAE promotes RNase 7 expression by inhibiting late-phase autophagy.** (**A–F**) NHEKs were treated with RAE (0.1% [v/v]) for 72 h in the presence of resveratrol (50 μM), a SIRT1 activator. (**A**) Expression levels of RNase 7 mRNA were determined by RT-qPCR. (**B–F**) Protein expression levels of RNase 7 (B), phosphorylated ERK (C), LC3-II (D), p62 (E), and phosphorylated S6 (F) were analysed by western blotting. (**A–F**) Data represent the mean ± SD of three independent experiments. Different letters indicate significant differences based on post-hoc Tukey's test results. P < 0.05. Res: Resveratrol.

phosphorylated S6 were examined in RAE-induced RNase 7 expression. As a result, together with a significant increase in the protein expression levels of phosphorylated S6 following treatment with RAE alone, it was confirmed that the protein expression levels of

phosphorylated S6 following treatment with resveratrol alone (Fig 5F; Control vs. Res) and the combination of RAE and resveratrol (Fig 5F; Control vs. RAE+Res) were significantly decreased compared with control, implying that resveratrol accelerated early-phase autophagy under our experimental conditions.

From the above results, it can be concluded that the inhibition of late-phase autophagy (i.e., accumulation of autophagosomes) is required for RAE-induced RNase 7 expression.

## S6 is not a downstream target of extracellular signal-regulated kinase

In the present study, RAE treatment activated S6 in primary human keratinocytes. However, the relationship between S6 and ERK remains unclear. Therefore, we examined the protein expression levels of phosphorylated S6 in the presence of both RAE and PD98059, a MEK/ERK inhibitor. The protein expression levels of phosphorylated S6 in the presence of both RAE and PD98059 were not different from those in the presence of RAE alone (Fig 6; RAE vs. RAE+PD), suggesting that S6 is not a downstream target of ERK.

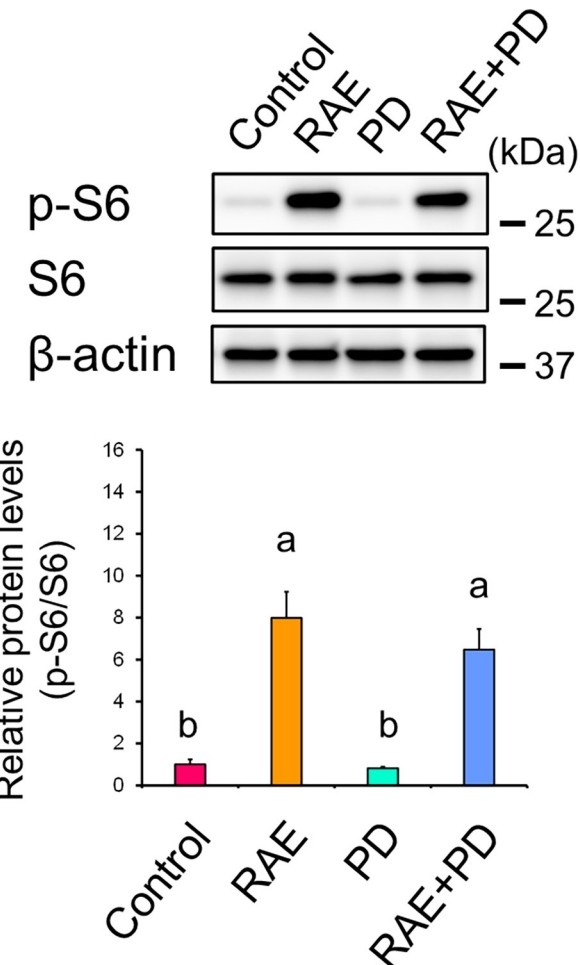

**Fig 6. ERK does not activate S6.** NHEKs were treated with RAE (0.1% [v/v]) for 72 h in the presence of PD98059 (30 μM), a MEK/ERK inhibitor. Protein expression levels of phosphorylated S6 were determined by western blotting. Data represent the mean ± SD of three independent experiments. Different letters indicate significant differences based on post-hoc Tukey's test results. P < 0.05. PD: PD98059.

### Isolation and identification of active compound present in *R. aculeatus* extract

We isolated and identified the active compound in RAE that induced RNase 7 expression, using column chromatography, preparative HPLC, and $^1$H and $^{13}$C NMR. $^1$H and $^{13}$C NMR analyses (Table 1) indicated that the active compound was spilacleoside, which was either spilacleoside A, spilacleoside B, or a mixture of these (Fig 7A) [38]. The concentration of spilacleoside in RAE was calculated to be approximately 83 μM (based on the preparative weight in each preparative process), indicating that the final concentration of spilacleoside when RAE was applied to the cells was 83 nM. Subsequently, using an ethanol solution (30%) containing the isolated spilacleoside (83 μM), the mRNA expression levels of RNase 7 (Fig 7B) and hBD-3 (Fig 7C) and the protein expression levels of RNase 7 (Fig 7D), LC3-II (Fig 7E), p62 (Fig 7F) and phosphorylated ERK (Fig 7G) were examined in human primary keratinocytes. The cells were treated with spilacleoside solution (0.1% [v/v]) for 72h. The results obtained following treatment with the spilacleoside solution were consistent with those obtained following RAE treatment. Hence, spilacleoside was confirmed to be the active compound in RAE.

## Discussion

In the present study, we investigated the effect of RAE on AMP expression and found that RAE primarily promoted RNase 7 expression in primary human keratinocytes. Since it was strongly suggested that autophagy may participate in the mechanism of RNase 7 expression induced by RAE (Fig 3), we focused our attention on examining the mechanism by which autophagy is involved. As demonstrated by the autophagic flux assay (Fig 3B and 3C), the increase in protein expression levels of p62 (Fig 3D) and the immunofluorescence analysis of colocalised LC3-II and p62 proteins (Fig 3E), RAE inhibited late-phase autophagy and caused the accumulation of autophagosomes. RAE-induced RNase 7 expression was promoted by ERK (Fig 2A, 2C and 2E), and activation of RNase 7 and ERK was inhibited by both EX-527 (Fig 4B–4D) and resveratrol (Fig 5A–5C). EX-527 suppressed early-phase autophagy (basal autophagic activity), which suggests that EX-527 inhibited the autophagosome formation activity of keratinocytes (Fig 4G), whereas resveratrol promoted it, which suggests that resveratrol finally eliminated the inhibition of late-phase autophagy induced by RAE treatment (the accumulation of autophagosomes) (Fig 5F). Therefore, the experiment with EX-527 indicated that the activity of early-phase autophagy is required for RAE-induced RNase 7 expression (Fig 4C). Conversely, the experiment with resveratrol indicated that the inhibition of late-phase autophagy is required for RAE-induced RNase 7 expression (Fig 5B). These results suggest that the accumulation of autophagosomes is crucial for RAE-induced RNase 7 expression. Moreover, both EX-527 and resveratrol would ultimately suppress autophagosome accumulation and simultaneously inhibited ERK activation. This suggests that ERK is a downstream target of autophagosome accumulation. Our findings suggest that RAE primarily promotes RNase 7 expression through the activation of ERK following the inhibition of late-phase autophagy (autophagosome accumulation). In addition, treatment with RAE and its mechanisms may provide a novel means of regulating AMP expression levels in human skin. Furthermore, RAE-induced RNase 7 expression indicates a new role of autophagy in human skin.

At present, the factors that activate ERK are unknown. Chloroquine, a late-phase autophagy inhibitor, has been reported to increase IL-23 production in a p38-dependent manner in monocyte-derived Langerhans-like cells [39]. In that report, it was proposed that impairment of the autophagic process (accumulation of autophagosomes) may result in p38 activation due to the inhibition of enzymatic degradation of upstream factors by autolysosomes or intracellular accumulation of autophagy-related substrates [39]. It is likely that ERK activation reported

**Table 1. ¹H-NMR and ¹³C-NMR spectral data of spilacleoside (600 MHz, pyridine-d5).**

| Carbon position | ¹H-NMR (ppm) | ¹³C-NMR (ppm) |
|---|---|---|
| 1 | 3.83 (1H, m) | 83.85 |
| 2 | 2.26 (2H, q, $J$ = 11.9Hz) | 37.9 |
| 3 | 3.88 (1H, br) | 68.2 |
| 4 | 2.71, 2.64 (2H, br) | 44.2 |
| 5 | - | 139.7 |
| 6 | 5.6 (1H, t, $J$ = 10Hz) | 125.2 |
| 7 | 1.8 (2H, m) | 32.2 |
| 8 | 1.49(1H, br) | 33.2 |
| 9 | 1.49(1H, br) | 50.5 |
| 10 | - | 43.1 |
| 11 | 2.86, 1.5 (2H, br) | 24.2 |
| 12 | 1.55, 1.26 (2H, br) | 40.7 |
| 13 | - | 41 |
| 14 | 1.08 (1H, d, $J$ = 7Hz) | 56.8 |
| 15 | 1.5, 1.08 (2H, br) | 32.6 |
| 16 | 4.8 (1H, m) | 83.2 |
| 17 | 4.01 (1H, m) | 61.7 |
| 18 | 0.94 (3H, s) | 17.1 |
| 19 | 1.36 (3H, s) | 15.2 |
| 20 | 2.87 (1H, t, $J$ = 6.7Hz) | 37.7 |
| 21 | 1.08 (3H, d, $J$ = 7Hz) | 15.1 |
| 22 | - | 111.9 |
| 23 | 4.01 (1H, m) | 61.7 |
| 24 | 4.8 (1H, d, $J$ = 3.9Hz) | 82.8 |
| 25 | - | 143.9 |
| 26 | 4.01, 4.83 (2H, d, $J$ = 12Hz) | 61.6 |
| 27 | 5.2, 5.1 (2H, s) | 114.5 |
| 1' | 4.64 (1H, d, $J$ = 7.7Hz) | 100.2 |
| 2' | 4.49(1H, t, $J$ = 8.4Hz) | 74.7 |
| 3' | 4.1 (1H, br) | 76.1 |
| 4' | 3.98 (1H, m) | 70.3 |
| 5' | 4.27, 3.64 (2H, d, $J$ = 12Hz) | 68.1 |
| 1" | 6.2 (1H, br) | 97.9 |
| 2" | 5.99 (1H, br) | 70.7 |
| 3" | 5.96 (1H, m) | 70.6 |
| 4" | 5.66 (1H, t, $J$ = 10Hz) | 72.1 |
| 5" | 5 (1H, m) | 66.7 |
| 6" | 1.43 (3H, d, $J$ = 6.2Hz) | 18.3 |
| AcO | 1.94 (3H, s) | 20.9 |
|  | - | 170.7 |
|  | 2.01 (3H, s) | 21 |
|  | - | 170.7 |
|  | 2.13 (3H, s) | 21 |
|  | - | 170.6 |
| 1‴ | 5.36 (1H, d, $J$ = 7.8) | 105.6 |
| 2‴ | 4.28 (1H, m) | 74.8 |
| 3‴ | 4.28 (1H, m) | 76.2 |

(*Continued*)

**Table 1.** (Continued)

| Carbon position | ¹H-NMR (ppm) | ¹³C-NMR (ppm) |
|---|---|---|
| 4‴ | - | 108.5 |
| 5‴ | 3.98 (1H, m) | 73.3 |
| 6‴ | 1.33 (3H, d, *J* = 6.3Hz) | 12.9 |
| 1‴' | - | 171.7 |
| 2‴' | 4.75 (1H, s) | 80.4 |
| 3‴' | - | 74.9 |
| 4‴' | 1.9 (2H, m) | 33.4 |
| 5‴' | 1.59 (3H, m) | 20.6 |
| 6‴' | 0.99 (3H, s) | 8.9 |

Carbon positions were indicated using spilacleoside A as a representative structure (Fig 7A). *J*, coupling constant expressed in Hz; s, singlet; br, broad; d, doublet; t, triplet; m, multiple; H, proton.

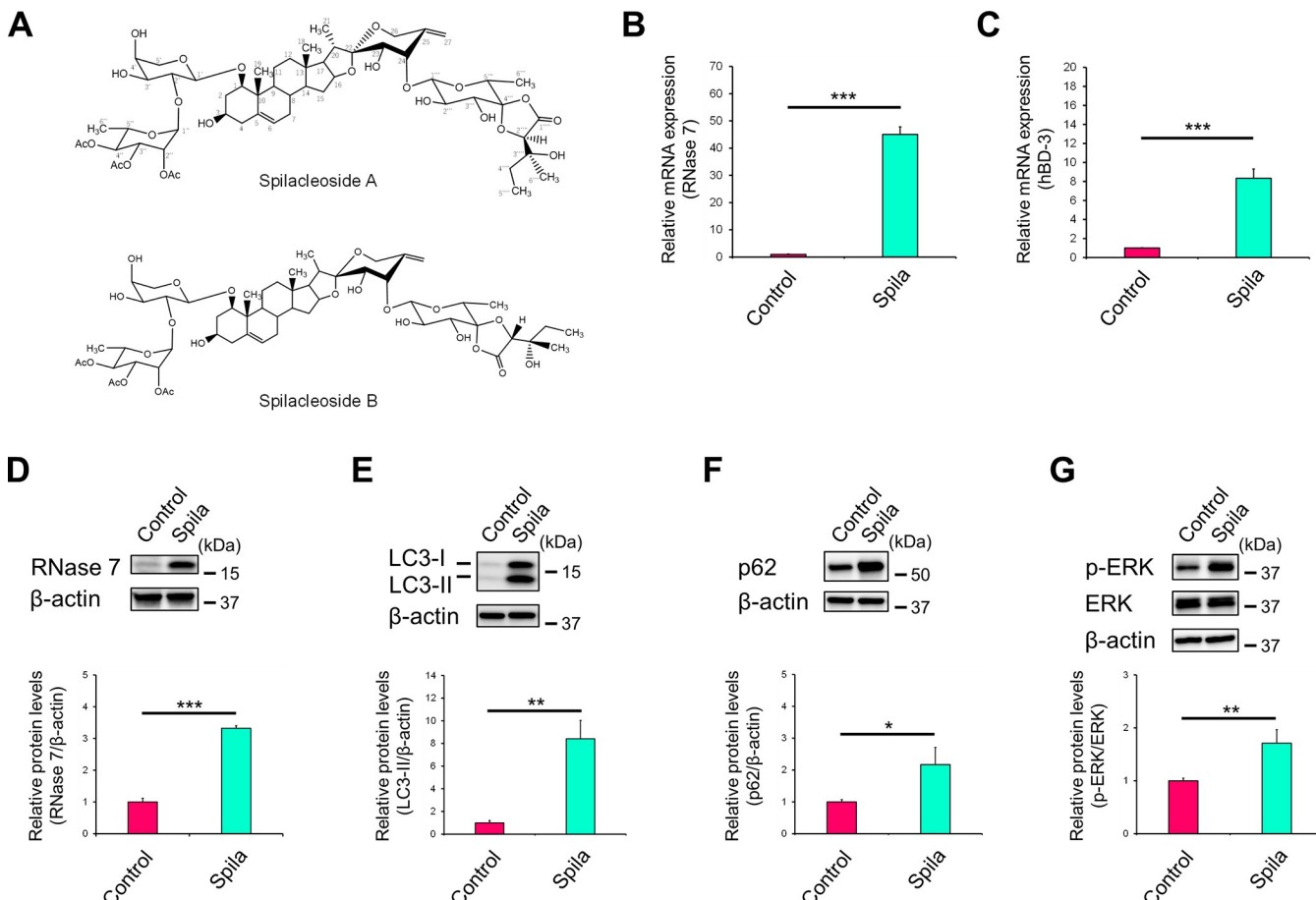

**Fig 7. Spilacleoside is an active compound in RAE.** (**A**) Chemical structure of spilacleoside A and B. (**B–G**) NHEKs were treated with spilacleoside (83 nM) for 72 h. (**B, C**) mRNA expression levels of RNase 7 (B) and hBD-3 (C) were determined by RT-qPCR. (**D–G**) Protein expression levels of RNase 7 (D), LC3-II (E), p62 (F), and phosphorylated ERK (G) were analysed by western blotting. (**B–G**) Data represent the mean ± SD of three independent experiments. Student's *t*-test. * P < 0.05, ** P < 0.01, ***P < 0.001. Spila: Spilacleoside.

in the present study might have occurred via mechanisms similar to those mentioned in the previous report [36]. In addition, unlike the scenario of phosphorylated ERK, EX-527 or resveratrol, which ultimately suppress autophagosome accumulation, did not affect the RAE-induced increase in protein expression levels of LC3-II and p62 proteins, which are autophagy-related substrates. Therefore, neither LC3-II nor p62 appear to be central regulators of ERK activation.

The absence of RAE-induced S6 activation in the presence of EX-527 and resveratrol suggests the potential involvement of S6 activation in the pathway via which RAE induces RNase 7 expression. S6 activation is known to be regulated by the mTOR/ribosomal S6 kinase (S6K) and the ERK/p90 ribosomal S6 kinase (RSK) pathways [40]. In this study, S6 activation was not inhibited in the presence of PD98059, a MEK/ERK inhibitor (Fig 6), suggesting that S6 was not a downstream target of ERK, and the ERK/RSK pathway is not involved in RAE-induced RNase 7 expression. Additionally, it was found that the activity of early-phase autophagy was required for RAE-induced RNase 7 expression. This implies that mTOR is not activated in RAE-induced RNase 7 expression and the mTOR/S6K pathway is also not involved in it. These findings suggest that S6 is not activated by the mTOR/S6K and/or ERK/RSK pathways; it seems that S6 is not the direct (key) factor in the pathway via which RAE induces RNase 7 expression. Thus, in addition to ERK activation, S6 activation might also occur through autophagosome accumulation. However, at present, the role of S6 in RAE-induced RNase 7 expression is unknown.

Even though mTOR is not activated in RAE-induced RNase 7 expression, the protein expression levels of phosphorylated S6 were not inversely correlated to those of LC3-II. When the protein expression levels of LC3-II are increased, S6 activation is expected to be decreased. In the present study, all protein expression levels of LC3-II, p62 and phosphorylated S6 were increased by RAE treatment. In line with our results, activation of Rubicon, an inhibitor of late-phase autophagy, has been reported to increase the expression levels of LC3-II and phosphorylated S6 in a manner inconsistent with their protein expression regulated by the mTOR-dependent autophagy pathway [41]. Simultaneously, Rubicon activation also increased the expression levels of p62 [41]. This phenomenon may be commonly occurring when late-phase autophagy is inhibited. In such cases, the expression of LC3-II, p62 and phosphorylated S6 might be regulated in part by early-phase autophagy via mTOR-dependent autophagy pathway and, for the most part, by unknown mTOR-independent mechanisms. Moreover, this consideration might explain why the expression levels of LC3-II and p62 proteins were not influenced in the presence of EX-527 or resveratrol. The mechanism by which RAE induces RNase 7 expression was not fully elucidated in this study. Therefore, further studies including knockdown experiments of genes such as ULK1, ATG 7, and SIRT1, will be needed to better understand RNase 7 expression induced by RAE treatment.

In the present study, we found that RAE, known for its high safety level, induced the expression of RNase 7 without affecting cell viability in primary human keratinocytes. Stimulation of primary human keratinocytes with a mixture of five inflammatory cytokines, including TNF, IL-1α, IL-17A, IL-22, and oncostatin M, increases the mRNA expression levels of S100A7, hBD-2, and LL-37 [42]. However, in our study, RAE treatment did not affect the mRNA expression levels of S100A7 and hBD-2. In addition, the mRNA expression levels of LL-37 were significantly increased by RAE treatment, but the increase was negligible. Furthermore, it is widely acknowledged that ERK (i.e., phosphorylated ERK) is not necessarily an inflammatory mediator. Therefore, our findings suggest that the expression of AMPs such as RNase 7, which was induced by RAE treatment, might occur in noninflammatory conditions. However, further investigations are warranted to verify that this process does not occur under inflammatory conditions in human skin.

It has been suggested that autophagy is constitutively active in the epidermal granular layer [43] and that during terminal differentiation, keratinocytes lose intercellular organelles, including nuclei, via autophagy [43,44], indicating that autophagy regulates the terminal differentiation of keratinocytes. In contrast, Daugelaviciene et al. demonstrated that, despite the impairment of the function of lysosomes, which is essential for autophagy, photodynamic treatment induces the terminal differentiation of keratinocytes [45]. In addition, even if AMP-activated protein kinase (AMPK), which induces autophagy by inhibiting mTOR, is genetically deleted in epidermal keratinocytes, the terminal differentiation of keratinocytes is unaffected [46]. Overall, the role of autophagy in the terminal differentiation of keratinocytes is controversial. During the terminal differentiation, keratinocytes in the granular layer begin to lose their nuclei and organelles and form a cornified layer. This implies that autophagy is also presumed to cease with the terminal differentiation of keratinocytes, as keratinocytes which lost a nuclei and organelles are no longer viable. In human skin, AMPs such as LL-37, hBD-2, and hBD-3 are stored in lamellar bodies and released through these vesicles into the extracellular spaces of the stratum corneum during the terminal differentiation of keratinocytes [47–49]. This suggests that autophagy, which is a degrading process, is not required for releasing AMPs to the extracellular spaces of the stratum corneum at the end of the terminal differentiation of keratinocyte. If autophagy is constantly active during the terminal differentiation of keratinocytes, the AMPs produced in keratinocytes of the granular layer will be broken down by autophagy. In this study, we reported that RNase 7 expression was induced by inhibition of late-phase autophagy, although the results were obtained in the presence of RAE. Therefore, we speculated that the expression of certain AMPs such as RNase 7 is induced along with the cessation of autophagy during the terminal differentiation of keratinocytes. That is, the terminal differentiation of keratinocytes under physiological conditions does not require constant autophagy, and it is at least partially related to the expression of RNase 7 under physiological conditions. Thus, the RAE used in the present study might have accelerated RNase 7 expression that is induced by the cessation of autophagy during the terminal differentiation of keratinocytes under physiological conditions. However, it remains uncertain whether RNase 7 expression is induced by the terminal differentiation of keratinocytes under physiological conditions and if there are specific factors that stop or inhibit late-phase autophagy under physiological conditions.

In the autophagic flux assay with HCQ and BA1, we showed that RAE inhibited late-phase autophagy in RNase 7 expression induced by RAE treatment. The protein expression levels of LC3-II induced by RAE in the presence of HCQ or BA1 were significantly decreased compared with those with RAE alone. This suggests that the inhibitory effect of RAE was weakened by them. In other words, it implies that the active compound (spilacleoside) in RAE inhibits late-phase autophagy at least in part in a manner similar to HCQ and BA1. Moreover, because the concentration of spilacleoside in RAE (83 nM) is much lower than HCQ (10 μM) and almost equal to BA1 (50 nM), its activity as a late-phase autophagy inhibitor might be quite high. Therefore, spilacleoside might become a novel lead compound for the development of highly effective autophagy inhibitors. It is well known that the mechanisms by which HCQ [50] and BA1 [51] inhibit late-phase autophagy are different, and the structures of these three compounds are also much different. The elucidation of functional mechanisms of spilacleoside as a late-phase autophagy inhibitor will be one of the most important issues in the future.

Given that autophagy is a housekeeping system in cells, it is thought to play a pivotal role in the prevention of various diseases such as cancer, neurodegenerative disorders, diabetes, obesity, inflammation, heart disease, and ageing [52]. Therefore, it is widely recognized that the activation of autophagy and/or the maintenance of autophagic activity are necessary to promote and maintain good health in daily life. However, it has recently been found that excessive

autophagy (or abnormally activated autophagy) is not beneficial for the treatment of diseases and human health. The following examples of excessive autophagy have been reported in the field of dermatology: autophagy-based unconventional secretion of high-mobility group box 1 is involved in psoriatic skin inflammation [42]; autophagy activated by advanced glycation end products causes refractory wounds by promoting M1 polarisation of macrophages in patients with diabetes [53]. In the present study, we showed that spilacleoside is not only an AMP inducer but also an effective autophagy inhibitor. Thus, spilacleoside and RAE containing it may be useful for inhibiting excessive autophagy as described above.

In summary, we have demonstrated that RAE, which contains spilacleoside as an active compound, promotes the expression of AMPs, in particular RNase 7, through ERK activation following the inhibition of late-phase autophagy in primary human keratinocytes. Moreover, these findings provide new insights into the means of regulating the expression levels of beneficial AMPs such as RNase 7, which may be applied directly to healthy skin, and the role of autophagy in human skin.

## Supporting information

**S1 File. Raw data.**
(XLSX)

**S1 Fig. Raw images of immunoblotting.**
(PPTX)

## Acknowledgments

The authors thank Atsuko Otsuka (Biological Science Research, Kao Corporation, Japan) for technical assistance.

## Author Contributions

**Conceptualization:** Shigeyuki Ono.

**Formal analysis:** Shigeyuki Ono, Akiko Kawasaki, Kotaro Tamura.

**Investigation:** Shigeyuki Ono, Akiko Kawasaki, Kotaro Tamura.

**Methodology:** Shigeyuki Ono, Akiko Kawasaki.

**Supervision:** Yoshihiko Minegishi, Takuya Mori, Noriyasu Ota.

**Visualization:** Shigeyuki Ono.

**Writing – original draft:** Shigeyuki Ono.

**Writing – review & editing:** Shigeyuki Ono, Yoshihiko Minegishi.

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
