## [Decision Letter · Decision Letter 0]

7 Jun 2024

PONE-D-24-15816

Ruscus aculeatus extract promotes RNase 7 expression through ERK activation following inhibition of late-phase autophagy in primary human keratinocytes.

PLOS ONE

Dear Dr. Ono,

Thank you for submitting your manuscript to PLOS ONE. After careful consideration, we feel that it has merit but does not fully meet PLOS ONE’s publication criteria as it currently stands. Therefore, we invite you to submit a revised version of the manuscript that addresses the points raised during the review process.

We look forward to receiving your revised manuscript.

Kind regards,

Ravikanth Nanduri, Ph. D.

Academic Editor

PLOS ONE

Journal Requirements:

Additional Editor Comments:

Based on both the reviewers, this manuscript requires several additional experiments to be accepted in PlosOne.

Please answer all the comments of the reviewers through additional experiments to support the hypothesis.

Reviewers' comments:

Reviewer's Responses to Questions

**Comments to the Author**

1. Is the manuscript technically sound, and do the data support the conclusions?

Reviewer #1: Partly

Reviewer #2: Yes

2. Has the statistical analysis been performed appropriately and rigorously? 

Reviewer #1: Yes

Reviewer #2: Yes

3. Have the authors made all data underlying the findings in their manuscript fully available?

Reviewer #1: Yes

Reviewer #2: Yes

4. Is the manuscript presented in an intelligible fashion and written in standard English?

Reviewer #1: No

Reviewer #2: Yes

5. Review Comments to the Author

Reviewer #1: The Manuscript entitled: Ruscus aculeatus extract promotes RNase 7 expression through ERK activation following inhibition of late-phase autophagy in primary human keratinocytes is interesting

But has quite a few important experiments and explanations to consider.

Major comments:

1) Fig3D: LC3 punctae are not visible in the figures. To clarify the claim of early and late phase autophagy authors are requested to provide mRFP-GFPLC3 data.

Authors should provide the better images or repeat with LC3 and p62 punctae.

2) Fig 4A: authors are requested to provide the immunoblots to prove the points with wortmanin like provided with EX527, because here authors are claiming to show the outcome of inhibition of early and late phase autophagy on RNAase 7 expression including p-ERK, also authors are requested to provide the evidence of early autophagy genes ULK1 and PIK3C3 KD and the effect on all the proteins LC3, p62, p-ERK and RNAase7, similary authors are requested to conduct late phase autophagy gene KD ATG7 which effects LC3 lipidation and provide the evidence for the effect on RNAase7.

3) Fig 4: Authors have mentioned EX-527 as mTOR activators although literature search shows ERX as mTOR suppressor (which means autophagy activator), authors are requested to clarify and rectify the claim here, otherwise it does not conclude what authors are trying to show here. Authors are claiming RAE as late phase autophagy inhibitor but p-S6 show it is activator or mTOR which controls the early autophagy, therefore authors own results reject the claim of inhibition of late phase autophagy, although it can be seen autophagy regulates the expression of RNAase7 expression.

4) figure 6 alone with one immunoblot is not required, although reviewer understands it is different from other figures but to make it a separate figure more data is required.

Minor comments:

1) authors are requested to keep blots for p62 and LC3 together in every figures.

2) all the blots specifically where the difference is there as per authors claim

Reviewer #2: In the manuscript titled "Ruscus aculeatus extract (RAE) promotes RNase 7 expression through ERK activation following inhibition of late-phase autophagy in primary human keratinocytes," the author explores the impact of RAE on AMP expression. The study reveals that RAE predominantly enhances RNase 7 expression in primary human keratinocytes by activating ERK signaling subsequent to late-phase autophagy inhibition. Moreover, the author underscores the significance of autophagy initiation for RNase 7 expression, thus elucidating the critical role of autophagosome formation and accumulation in mediating RAE-induced upregulation of RNase 7. Below are my comments:

Acknowledgment of Manuscript Quality:

"The manuscript was effectively written, with data presented in a clear and concise manner."

Recognition of Experimental Design:

"The experimental design was robust, and the data appeared organized and well-presented."

Appreciation for Statistical Analysis:

"I appreciate the author's use of valid statistical analysis tests, which bolster the credibility of the findings."

Suggestion for Experimentation Expansion:

"While HCQ was appropriately used to inhibit phagolysosome formation, I recommend the exploration of additional late-phase autophagy inhibitors, such as Bafilomycin A1. This broader experimentation would enhance the comprehensiveness of the study."

"Similarly all of the experiments were performed on a single cell line, it would be more informative if the author provided data from a second cell line"

Request for Image Clarity:

"I suggest improving the image quality of the colocalization experiments in Figure 3. In the current representations, it was challenging to discern the localization of LC3 and p62, which are pivotal for accurate interpretation."

6. PLOS authors have the option to publish the peer review history of their article (what does this mean?). If published, this will include your full peer review and any attached files.

Reviewer #1: **Yes: **Gaurav Sharma

Reviewer #2: **Yes: **Bhagyaraj Ella

---

## [Decision Letter · Decision Letter 1]

21 Oct 2024

PONE-D-24-15816R1Ruscus aculeatus extract promotes RNase 7 expression through ERK activation following inhibition of late-phase autophagy in primary human keratinocytes.PLOS ONE

Dear Dr. Ono,

Thank you for submitting your revised manuscript to PLOS ONE. The manuscript was substantially improved. However, after careful consideration, we feel that it has merit but does not fully meet PLOS ONE’s publication criteria as it currently stands. Therefore, we invite you to submit a revised version of the manuscript that addresses the points raised by the new Academic Editor during the second round of the review process.

We look forward to receiving your revised manuscript.

Kind regards,

Michel Simon, Ph. D.

Academic Editor

PLOS ONE

Journal Requirements:

Additional Editor Comments:

The manuscript was improved.

My main point concerns the inhibition/activation of early autophagy and effects of RAE on RNase7 expression. Indeed, reduced detection of RNase7 was observed after inhibition of early autophagy. I do not understand why a down-regulation of RNase7 should be expected after activation of early autophagy; I rather suspect an increased expression. Please clarify this point.

Some minor issues also appeared to this Editor, as follows.

1) Please explain the differences between letters a-d in Figures 2-6.

2) The entire blot corresponding to pp38 detection is of very poor quality with many un-specific bands. Please provide a blot of better quality or delete the data related to p38 (Fig2D).

3) N=3; does this mean 3 technical replicates (same extract) or biological replicates (3 different cultures)? Please clarify.

Reviewers' comments:

Reviewer's Responses to Questions

**Comments to the Author**

1. If the authors have adequately addressed your comments raised in a previous round of review and you feel that this manuscript is now acceptable for publication, you may indicate that here to bypass the “Comments to the Author” section, enter your conflict of interest statement in the “Confidential to Editor” section, and submit your "Accept" recommendation.

Reviewer #2: All comments have been addressed

2. Is the manuscript technically sound, and do the data support the conclusions?

Reviewer #2: Yes

3. Has the statistical analysis been performed appropriately and rigorously? 

Reviewer #2: Yes

4. Have the authors made all data underlying the findings in their manuscript fully available?

Reviewer #2: Yes

5. Is the manuscript presented in an intelligible fashion and written in standard English?

Reviewer #2: Yes

6. Review Comments to the Author

Reviewer #2: Author addressed all the previous comments and provide additional data that strengthens their findings.

7. PLOS authors have the option to publish the peer review history of their article (what does this mean?). If published, this will include your full peer review and any attached files.

Reviewer #2: **Yes: **Bhagyaraj Ella

---

## [Editor Report · Decision Letter 2]

19 Nov 2024

Ruscus aculeatus extract promotes RNase 7 expression through ERK activation following inhibition of late-phase autophagy in primary human keratinocytes.

PONE-D-24-15816R2

Dear Dr. Ono,

We’re pleased to inform you that your manuscript has been judged scientifically suitable for publication and will be formally accepted for publication once it meets all outstanding technical requirements. In addition, since the authors could not provide a blot of better quality corresponding to pp38 detection, the data related to p38 must not be mentioned at all. Indeed, no conclusion can be deduced from them. They are not necessary for the interpretation of the study.

Kind regards,

Michel Simon, Ph. D.

Academic Editor

PLOS ONE
---

## [Editor Report · Acceptance letter]

21 Nov 2024

PONE-D-24-15816R2 

PLOS ONE

Dear Dr. Ono, 

I'm pleased to inform you that your manuscript has been deemed suitable for publication in PLOS ONE. Congratulations! Your manuscript is now being handed over to our production team.

Kind regards, 

on behalf of

Dr. Michel Simon 

Academic Editor

PLOS ONE